# Identifiable Deep Generative Models via Sparse Decoding

**Gemma E. Moran**                                                          *gm2918@columbia.edu*
*Columbia University*

**Dhanya Sridhar**
*Mila - Quebec AI Institute and Université de Montréal*

**Yixin Wang**
*University of Michigan*

**David M. Blei**
*Columbia University*

**Reviewed on OpenReview:** *https://openreview.net/forum?id=vd0onGWZbE*

## Abstract

We develop the sparse VAE for unsupervised representation learning on high-dimensional data. The sparse VAE learns a set of latent factors (representations) which summarize the associations in the observed data features. The underlying model is sparse in that each observed feature (i.e. each dimension of the data) depends on a small subset of the latent factors. As examples, in ratings data each movie is only described by a few genres; in text data each word is only applicable to a few topics; in genomics, each gene is active in only a few biological processes. We prove such sparse deep generative models are identifiable: with infinite data, the true model parameters can be learned. (In contrast, most deep generative models are not identifiable.) We empirically study the sparse VAE with both simulated and real data. We find that it recovers meaningful latent factors and has smaller heldout reconstruction error than related methods.

## 1 Introduction

In many domains, high-dimensional data exhibits variability that can be summarized by low-dimensional latent representations, or factors. These factors can be useful in a variety of tasks including prediction, transfer learning, and domain adaptation (Bengio et al., 2013).

To learn such factors, many researchers fit deep generative models (DGMs) (Kingma and Welling, 2014; Rezende et al., 2014). A DGM models each data point with a latent low-dimensional representation (its factors), which is passed through a neural network to generate the observed features. Given a dataset, a DGM can be fit with a variational autoencoder (VAE), a method that provides both the fitted parameters to the neural network and the approximate posterior factors for each data point.

In this paper, we make two related contributions to the study of deep generative models.

First, we develop the sparse DGM. This model is a DGM where each observed feature only depends on a subset of the factors – that is, the mapping from factors to features is sparse. This notion of sparsity often reflects the type of underlying patterns that we anticipate finding in real-world data. In genomics, each gene is associated with a few biological processes; in text, each term is applicable to a few underlying topics; in movie ratings, each movie is associated with a few genres.

In detail, the model implements sparsity through a per-feature masking variable. When generating a data point, this mask is applied to the latent representation before producing each feature. In practice, we learn the per-feature mask with the help of the Spike-and-Slab Lasso prior (Ročková and George, 2018),

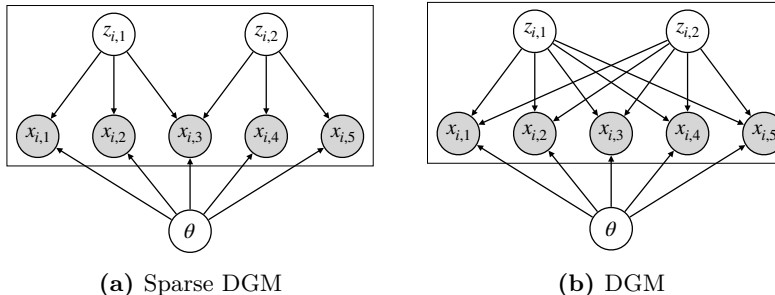

**(a)** Sparse DGM          **(b)** DGM

**Figure 1:** In a DGM, a feature $x_{ij}$ depends on all factors, $z_{ik}$. A sparse DGM is displayed where features $x_{i1}, x_{i2}$ depend only on $z_{i1}$; $x_{i3}$ depends on $z_{i1}$ and $z_{i2}$; and $x_{i4}, x_{i5}$ depend only on $z_{i2}$. The features are passed through the same neural network $f_\theta$.

a sparsity-inducing prior which enjoys good properties in more classical settings. We fit this sparse DGM and SSL prior with amortized variational inference (Gershman and Goodman, 2014; Kingma and Welling, 2014). We call this procedure the sparse VAE.

Second, we study identifiability in sparse DGMs. A model is identifiable if each setting of its parameters produces a unique distribution of the observed data. With an unidentifiable DGM, even with infinite data from the model, we cannot distinguish the true factors. Identifiable factor models are important to tasks such as domain adaptation (Locatello et al., 2020), transfer learning (Dittadi et al., 2021), and fair classification (Locatello et al., 2019a). Most DGMs are not identifiable (Locatello et al., 2019b).

Specifically, we prove that a sparse DGM is identifiable if each latent factor has at least two "anchor features." An anchor feature is a dimension of the data which depends on only one latent factor. For example, in text data, an anchor feature is a word that is applicable to only one underlying topic (Arora et al., 2013). Note that the anchor features do not need to be known in advance for our results to hold; we only need to assume they exist. Further, we do not need to assume that the true factors are drawn independently; in many applications, such an independence assumption is unrealistic (Träuble et al., 2021).

The sparse VAE and the anchor assumption are related in that the SSL prior will likely yield the kind of sparse mapping that satisfies the anchor assumption. What this means is that if the data comes from an identifiable DGM, i.e., one that satisfies the anchor assumption, then the SSL prior will encourage the sparse VAE to more quickly find a good solution. In this sense, this paper is in the spirit of research in Bayesian statistics, which establishes theoretical conditions on identifiability and then produces priors that favor identifiable distributions.

Empirically, we compare the sparse VAE to existing algorithms for fitting DGMs: the VAE (Kingma and Welling, 2014), $\beta$-VAE (Higgins et al., 2017), Variational Sparse Coding (VSC, Tonolini et al., 2020), and OI-VAE (Ainsworth et al., 2018). We find that (i) on synthetic data, the sparse VAE recovers ground truth factors, including when the true factors are correlated; (ii) on text and movie-ratings data, the sparse VAE achieves better heldout predictive performance; (iii) on semi-synthetic data, the sparse VAE has better heldout predictive performance on test data that is distributed differently to training data; (iv) the sparse VAE finds interpretable structure in text, movie-ratings and genomics data.

In summary, the contributions of our paper are as follows:

- We develop a sparse DGM which has Spike-and-Slab Lasso priors (Ročková and George, 2018) and develop an algorithm for fitting this model, the sparse VAE.

- We prove that sparse DGMs are identified under an anchor feature assumption.

- We study the sparse VAE empirically. It outperforms existing methods on synthetic, semi-synthetic, and real datasets of text, ratings, and genomics.

**Related Work.** The sparse DGM provides a contribution to sparse methods in linear and nonlinear representation learning. The sparse DGM has a sparsity-inducing prior on the factor-to-feature mapping

(i.e., the decoder). Similar priors have also been applied in linear factor analysis to induce sparsity in the factor-to-feature mapping (i.e., the loadings matrix); see, for example Bernardo et al. (2003); Bhattacharya and Dunson (2011); Carvalho et al. (2008); Knowles and Ghahramani (2011); Ročková and George (2016).

Beyond linearity, Barello et al. (2018) considers a hybrid linear/nonlinear VAE which uses a sparse linear decoder and a neural network encoder. In nonlinear representation learning, Tonolini et al. (2020) imposes sparsity-inducing priors directly on the latent factors. Instead, the sparse DGM of this paper involves a sparsity-inducing prior on the factor-to-feature mapping. Rhodes and Lee (2021) indirectly induces sparsity in the factor-to-feature mapping of their DGM via $L_1$ regularization of the mapping's Jacobian.

The sparse VAE closely relates to the OI-VAE (Ainsworth et al., 2018), a method for modeling grouped features where the factor-to-group mapping is sparse. However, there are several differences. First, the OI-VAE uses a Group Lasso prior while the sparse VAE has a Spike-and-Slab Lasso prior (SSL, Ročková and George, 2018). The SSL prior mitigates common issues with the Lasso prior, including under-regularization of small coefficients and over-regularization of large coefficients (Ghosh et al., 2016), and a sub-optimal posterior contraction rate (Castillo et al., 2015). Second, we study the identifiability of sparse DGMs, a property that Ainsworth et al. (2018) does not investigate.

This paper also contributes to the literature that uses the anchor feature assumption for identifiability. In linear models, the anchor feature assumption has led to identifiability results for topic models (Arora et al., 2013), non-negative matrix factorization (Donoho and Stodden, 2003), and linear factor analysis (Bing et al., 2020). The key idea is that the anchor assumption removes the rotational invariance of the latent factors. This idea also arises in identifiable factor analysis (Rohe and Zeng, 2020) and independent component analysis, where rotational invariance is removed by assuming that the components are non-Gaussian (Eriksson and Koivunen, 2004).

Finally, this paper is related to methods in identifiable nonlinear representation learning. To achieve identifiability, most methods require weak supervision: Khemakhem et al. (2020); Mita et al. (2021); Zhou and Wei (2020) require auxiliary data; Locatello et al. (2020) requires paired samples; Kügelgen et al. (2021) relies on data augmentation; and Hälvä et al. (2021); Ahuja et al. (2022); Lachapelle et al. (2022) leverage known temporal or spatial dependencies among the samples. In contrast, the sparse DGM does not need auxiliary information or weak supervision for identifiability; instead, the anchor feature assumption is sufficient. Relatedly, Horan et al. (2021) achieves identifiable representations under a "local isometry" assumption, meaning a small change in a factor corresponds to a small change in the features. In contrast to this paper, however, Horan et al. (2021) requires the factors to be independent; we do not need such an independence assumption.

## 2 The Sparse Variational Autoencoder

The observed data is a vector of $G$ features $\boldsymbol{x}_i \in \mathbb{R}^G$ for $i \in \{1, \ldots, N\}$ data points. The goal is to estimate a low-dimensional set of factors $\boldsymbol{z}_i \in \mathbb{R}^K$ where $K \ll G$.

In a standard DGM, the vector $\boldsymbol{z}_i$ is a latent variable that is fed through a neural network to reconstruct the distribution of $\boldsymbol{x}_i$. The sparse DGM introduces an additional parameter $\boldsymbol{w}_j \in \mathbb{R}^K$, $j \in \{1, \ldots, G\}$, a sparse per-feature vector that selects which of the $K$ latent factors are used to produce the $j$th feature of $\boldsymbol{x}_i$. Further, the sparse DGM models the prior covariance between factors with the positive definite matrix $\boldsymbol{\Sigma}_z \in \mathbb{R}^{K \times K}$. The sparse DGM is:

$$
\begin{aligned}
w_{jk} &\sim \text{Spike-and-Slab Lasso}(\lambda_0, \lambda_1, a, b), \quad k = 1, \ldots, K \\
\boldsymbol{z}_i &\sim \mathcal{N}_K(0, \boldsymbol{\Sigma}_z), \quad i = 1, \ldots, N \\
x_{ij} &\sim \mathcal{N}((f_\theta(\boldsymbol{w}_j \odot \boldsymbol{z}_i))_j, \sigma_j^2), \quad j = 1, \ldots, G
\end{aligned}
\tag{1}
$$

where $f_\theta : \mathbb{R}^K \to \mathbb{R}^G$ is a neural network with weights $\theta$, $\sigma_j^2$ is the per-feature noise variance and $\odot$ denotes element-wise multiplication. The Spike-and-Slab Lasso (Ročková and George, 2018) is a sparsity-inducing prior which we will describe below. (Any sparsity-inducing prior on $w_{jk}$ would be appropriate for a sparse DGM; we choose SSL because it enjoys theoretical guarantees and fast computation in more classical settings.)

---

**Algorithm 1:** The sparse VAE

---

**input**: data $\boldsymbol{X}$, hyperparameters $\lambda_0, \lambda_1, a, b, \boldsymbol{\Sigma}_z$
**output**: factor distributions $q_\phi(\boldsymbol{z}|\boldsymbol{x})$, selector matrix $\boldsymbol{W}$, parameters $\theta$
**while** *not converged* **do**

    **for** $j = 1, \ldots, G; \ k = 1, \ldots, K$ **do**
        Update
$$\mathbb{E}\left[\gamma_{jk}|w_{jk}, \eta_k\right] = \left[1 + \frac{1 - \eta_k}{\eta_k}\frac{\psi_0(w_{jk})}{\psi_1(w_{jk})}\right]^{-1};$$

    **for** $k = 1, \ldots, K$ **do**
        Update
$$\eta_k = \frac{\left(\sum_{j=1}^{G}\mathbb{E}\left[\gamma_{jk}|w_{jk}, \eta_k\right] + a - 1\right)}{a + b + G - 2};$$

    Update $\theta, \phi, \boldsymbol{W}$ with SGD according to Eq. 8;

---

In the data generating distribution in Eq. 1, the parameter $w_{jk}$ controls whether the distribution of $x_{ij}$ depends on factor $k$. If $w_{jk} \neq 0$, then $z_{ik}$ contributes to $x_{ij}$, while if $w_{jk} = 0$, $z_{ik}$ cannot contribute to $x_{ij}$. If $\boldsymbol{w}_j$ is sparse, then $x_{ij}$ depends on only a small set of factors. Such a sparse factor-to-feature relationship is shown in Figure 1.

Note that $x_{ij}$ depends on the same set of factors for every sample $i$. This dependency only makes sense when each $x_{ij}$ has a consistent meaning across samples. For example, in genomics data $x_{ij}$ corresponds to the same gene for all $i$ samples. (This dependency is not reasonable for image data, where pixel $j$ has no consistent meaning across samples.)

**Spike-and-Slab Lasso.** The parameter $w_{jk}$ has a SSL prior (Ročková and George, 2018), which is defined as a hierarchical model,

$$\eta_k \sim \text{Beta}(a, b), \tag{2}$$

$$\gamma_{jk} \sim \text{Bernoulli}(\eta_k), \tag{3}$$

$$w_{jk} \sim \gamma_{jk}\psi_1(w_{jk}) + (1 - \gamma_{jk})\psi_0(w_{jk}). \tag{4}$$

In this prior, $\psi_s(w) = \frac{\lambda_s}{2}\exp(-\lambda_s|w|)$ is the Laplace density and $\lambda_0 \gg \lambda_1$. The variable $w_{jk}$ is drawn *a priori* from either a Laplacian "spike" parameterized by $\lambda_0$, and is consequentially negligible, or a Laplacian "slab" parameterized by $\lambda_1$, and can be large.

Further, the variable $\gamma_{jk}$ is a binary indicator variable that determines whether $w_{jk}$ is negligible. The Beta-Bernoulli prior on $\gamma_{jk}$ allows for uncertainty in determining which factors contribute to each feature.

Finally, the parameter $\eta_k \in [0, 1]$ controls the proportion of features that depend on factor $k$. By allowing $\eta_k$ to vary, the sparse DGM allows each factor to contribute to different numbers of features. In movie-ratings data, for example, a factor corresponding to the action/adventure genre may be associated with more movies than a more esoteric genre.

Notice the prior on $\eta_k$ helps to "zero out" extraneous factor dimensions and consequently estimate the number of factors $K$. If the hyperparameters are set to $a \propto 1/K$ and $b = 1$, the Beta-Bernoulli prior corresponds to the finite Indian Buffet Process prior (Griffiths and Ghahramani, 2005).

### 2.1 Inference

In inference, we are given a dataset $\boldsymbol{x}_{1:N}$ and want to calculate the posterior $p(\theta, \boldsymbol{W}, \boldsymbol{\Gamma}, \boldsymbol{\eta}, \boldsymbol{z}_{1:N}|\boldsymbol{x}_{1:N})$. We will use approximate maximum *a posteriori* (MAP) estimation for $\boldsymbol{W}$, $\theta$ and $\boldsymbol{\eta}$, and amortized variational inference for $\boldsymbol{z}_{1:N}$. The full procedure for approximating this posterior is called a sparse VAE.

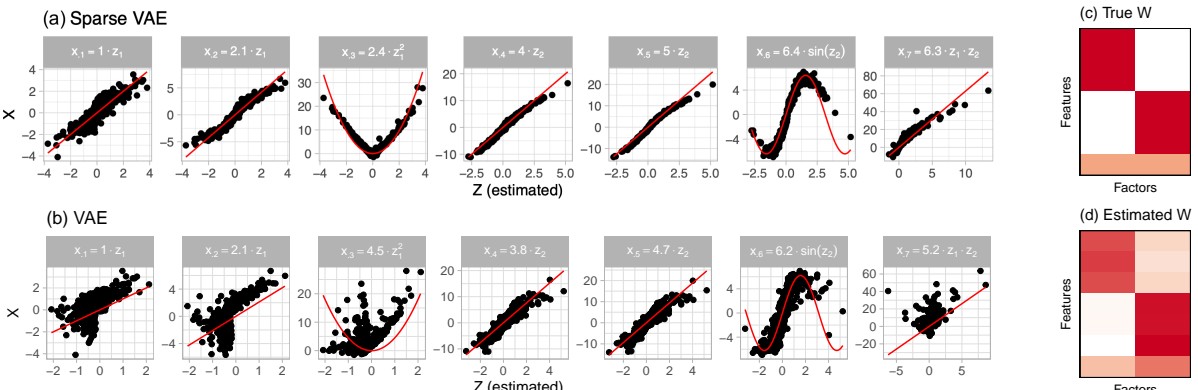

**Figure 2:** (a-b) The sparse VAE estimates factors which recover the true generative process; the VAE does not. The observed data is plotted against the estimated factors. The true factor-feature relationship is the red line; the best fit coefficients for the estimated factors are in the grey boxes. (c) The true $\boldsymbol{W}$ matrix. (d) The sparse VAE estimate of $\boldsymbol{W}$. (VAE has no $\boldsymbol{W}$ matrix).

The exact MAP objective is

$$\sum_{i=1}^{N} \log p_\theta(\boldsymbol{x}_i, \boldsymbol{W}, \boldsymbol{\eta}) = \sum_{i=1}^{N} \log \left[ \int p_\theta(\boldsymbol{x}_i | \boldsymbol{z}_i, \boldsymbol{W}) p(\boldsymbol{z}_i) d\boldsymbol{z}_i \right] + \log \int p(\boldsymbol{W}|\boldsymbol{\Gamma}) p(\boldsymbol{\Gamma}|\boldsymbol{\eta}) p(\boldsymbol{\eta}) d\boldsymbol{\Gamma}, \tag{5}$$

where $\boldsymbol{\Gamma} = \{\gamma_{jk}\}_{j,k=1}^{G,K}$ denotes the binary indicators. We lower bound Eq. 5 with variational approximations of the posterior of $\boldsymbol{z}_{1:N}$ and the exact posterior of the latent $\boldsymbol{\Gamma}$, which comes from the SSL prior. We approximate the posterior of the factors $p_\theta(\boldsymbol{z}_i|\boldsymbol{x}_i)$ by the variational family:

$$q_\phi(\boldsymbol{z}_i|\boldsymbol{x}_i) = \mathcal{N}_K(\mu_\phi(\boldsymbol{x}_i), \sigma_\phi^2(\boldsymbol{x}_i)), \tag{6}$$

where $\mu_\phi(\cdot)$ and $\sigma_\phi^2(\cdot)$ are neural networks with weights $\phi$, as in a standard VAE (Kingma and Welling, 2014; Rezende et al., 2014).

These approximations yield our final objective function:

$$\mathcal{L}(\theta, \phi, \boldsymbol{W}, \boldsymbol{\eta}) = \sum_{i=1}^{N} \left\{ \mathbb{E}_{q_\phi(\boldsymbol{z}_i|\boldsymbol{x}_i)}[\log p_\theta(\boldsymbol{x}_i|\boldsymbol{z}_i, \boldsymbol{W})] - D_{KL}(q_\phi(\boldsymbol{z}_i|\boldsymbol{x}_i)||p(\boldsymbol{z}_i)) \right\} \tag{7}$$

$$+ \mathbb{E}_{\boldsymbol{\Gamma}|\boldsymbol{W}, \boldsymbol{\eta}} \left[ \log[p(\boldsymbol{W}|\boldsymbol{\Gamma}) p(\boldsymbol{\Gamma}|\boldsymbol{\eta}) p(\boldsymbol{\eta})] \right]. \tag{8}$$

To optimize $\mathcal{L}(\theta, \phi, \boldsymbol{W}, \boldsymbol{\eta})$, we alternate between an expectation step and a maximization step. In the E-step, we calculate the complete conditional posterior of $\boldsymbol{\Gamma}$. In the M-step, we take gradient steps in model parameters and variational parameters with the help of reparameterization gradients (Kingma and Welling, 2014; Rezende et al., 2014). The sparse VAE procedure is in Algorithm 1.

The sparse VAE has a higher computational burden than a VAE. A VAE typically requires one forward pass through the neural network at each gradient step. In contrast, the sparse VAE requires $G$ forward passes. This additional computation is because for each of the $G$ features, the sparse VAE uses a different decoder input $\boldsymbol{w}_j \odot \boldsymbol{z}_i$, for $j = 1, \ldots, G$. (The VAE and sparse VAE optimize the same number of neural network parameters, however).

## 2.2 Synthetic example

For intuition, consider a synthetic dataset. We generate $N = 1000$ samples from a factor model, each with $G = 7$ features and $K = 2$ latent factors. We take the true factors to be correlated and generate them as $\boldsymbol{z}_i \sim N(0, \boldsymbol{C})$ where the true factor covariance $\boldsymbol{C}$ has diagonal entries $C_{kk} = 1$ and off-diagonal entries $C_{kk'} = 0.6$ for $k \neq k'$.

Given the factors, the data is generated as:

$$\boldsymbol{x}_i \sim \mathcal{N}_G(f(\boldsymbol{z}_i), \sigma^2 \boldsymbol{I}_G) \quad \text{with } \sigma^2 = 0.5. \tag{9}$$

The factor-to-feature mapping $f : \mathbb{R}^K \to \mathbb{R}^G$ is sparse:

$$f(\boldsymbol{z}_i) = (z_{i1}, 2z_{i1}, 3z_{i1}^2, 4z_{i2}, 5z_{i2}, 6\sin(z_{i2}), 7z_{i1} \cdot z_{i2}).$$

This mapping corresponds to the $\boldsymbol{W}$ matrix in Figure 2c.

We compare the sparse VAE with a VAE. For both, we initialize with an overestimate of the true number of factors ($K_0 = 5$). The sparse VAE finds factors that reflect the true generative model (Figure 2a). Moreover, the sparse VAE correctly finds the true number of factors by setting three columns of $\boldsymbol{W}$ to zero (Figure 2d). In contrast, the VAE recovers the second factor fairly well, but doesn't recover the first factor (Figure 2b).

We analyze this synthetic data further in § 4, as well as text, movie ratings and genomics data.

## 3 Identifiability of Sparse DGMs

For an identifiable model, given infinite data, it is possible to learn the true parameters. In both linear and nonlinear factor models, the factors are usually unidentifiable without additional constraints (Yalcin and Amemiya, 2001; Locatello et al., 2019b). This lack of identifiability means that the true factors $\boldsymbol{z}_i$ can have the same likelihood as another solution $\widetilde{\boldsymbol{z}}_i$. Even with infinite data, we cannot distinguish between these representations based on the likelihood alone. Further inductive biases or model constraints are necessary to narrow down the set of solutions.

In this section, we demonstrate that sparsity is a useful inductive bias for model identifiability. Consider the sparse deep generative model:

$$\begin{aligned} \boldsymbol{z}_i &\sim \mathcal{N}_K(0, \boldsymbol{C}), \quad i = 1, \dots, N \\ x_{ij} &\sim \mathcal{N}((f_\theta(\boldsymbol{w}_j \odot \boldsymbol{z}_i))_j, \sigma_j^2), \quad j = 1, \dots, G, \end{aligned} \tag{10}$$

where $\boldsymbol{C}$ is the true covariance matrix of the factors. Informally, if the masking variable $\boldsymbol{W}$ is sparse in a particular way, then we can identify the latent factors, up to coordinate-wise transformation. (Note: we only make an assumption about the sparse support of $\boldsymbol{W}$, not its distribution; the SSL prior in § 2 is a tool to expore the space of sparse $\boldsymbol{W}$ in the process of fitting the model.)

Specifically, we prove that for any solutions $\boldsymbol{z}$ and $\widetilde{\boldsymbol{z}}$ with equal likelihood, we have:

$$z_{ik} = g_k(\widetilde{z}_{ik}), \quad i = 1, \dots, N, \tag{11}$$

for all factors $k = 1, \dots, K$ (up to permutation of $k$), where $g_k : \mathbb{R} \to \mathbb{R}$ are invertible and differentiable functions.

This definition of identifiability is weaker than the canonical definition, for which we would need the two solutions $\boldsymbol{z}_i, \widetilde{\boldsymbol{z}}_i$ to be exactly equal. We use this weaker notion of identifiability because our goal is to isolate the dimensions of $\boldsymbol{z}_i$ which drive the observed response, and not necessarily find their exact value—i.e. we want to avoid mixing the dimensions of $\boldsymbol{z}_i$. For example, we want to be able to distinguish between solutions $\boldsymbol{z}$ and $\widetilde{\boldsymbol{z}} = \boldsymbol{P}\boldsymbol{z}$, for arbitrary rotation matrices $\boldsymbol{P}$.

**Starting point.** This paper builds on a body of work in identifiable linear unsupervised models (Arora et al., 2013; Bing et al., 2020; Donoho and Stodden, 2003). These works assume the existence of anchor features, features which depend on only one factor. The existence of anchor features helps with identifiability because they leave a detectable imprint in the covariance of the observed data.

We extend these results to the nonlinear setting. The key insight is that if two anchor features have the same nonlinear dependence on a factor, then we can apply results from the linear setting.

We first formally state the anchor feature assumption.

**A1. (Anchor feature assumption)** For every factor, $\boldsymbol{z}_{\cdot k}$, there are at least two features $\boldsymbol{x}_{\cdot j}, \boldsymbol{x}_{\cdot j'}$ which depend only on that factor. Moreover, the two features have the same mapping $f_j$ from the factors $\boldsymbol{z}_{\cdot k}$; that is, for all $i = 1, \ldots, N$:

$$\mathbb{E}[\boldsymbol{x}_{ij}|\boldsymbol{z}_i] = f_j(z_{ik}), \qquad \mathbb{E}[\boldsymbol{x}_{ij'}|\boldsymbol{z}_i] = f_j(z_{ik}). \tag{12}$$

We refer to such features as "anchor features."

The connection between the anchor feature assumption and the masking variable $\boldsymbol{W}$ is the following: If feature $j$ is an anchor for factor $k$, then $w_{jk} \neq 0$ and $w_{jk'} = 0$ for all $k \neq k'$.

**Roadmap.** We prove the identifiability of the sparse DGM in two parts. First, we prove that the latent factors are identifiable up to coordinate-wise transformation when the anchor features are known (Theorem 1). Second, we prove that the anchor features can be detected from the observed covariance matrix (Theorem 2). Together, Theorem 1 and Theorem 2 give the identifiability of the sparse DGM.

**Known anchor features.** The first result, Theorem 1, proves that the sparse DGM factors are identifiable if we are given the anchor features. If feature $j$ is an anchor feature for factor $k$, we set $w_{jk} = 1$ and $w_{jk'} = 0$ for all $k' \neq k$.

**Theorem 1.** *Suppose we have infinite data drawn from the model in Eq. 10 and A1 holds. Assume we are given the rows of $\boldsymbol{W}$ corresponding to the anchor features. Suppose we have two solutions with equal likelihood: $\{\widetilde{\theta}, \widetilde{\boldsymbol{z}}\}$ and $\{\widehat{\theta}, \widehat{\boldsymbol{z}}\}$, with*

$$p_{\widetilde{\theta}}(\boldsymbol{x}|\widetilde{\boldsymbol{z}}, \boldsymbol{W}) = p_{\widehat{\theta}}(\boldsymbol{x}|\widehat{\boldsymbol{z}}, \boldsymbol{W}). \tag{13}$$

*Then, the factors $\widetilde{\boldsymbol{z}}$ and $\widehat{\boldsymbol{z}}$ are equal up to coordinate wise transformations. For $g_k : \mathbb{R} \to \mathbb{R}$ and $i = 1, \ldots, N$,*

$$(\widetilde{z}_{i1}, \ldots, \widetilde{z}_{iK}) = (g_1(\widehat{z}_{i1}), \ldots, g_K(\widehat{z}_{iK})). \tag{14}$$

The proof of Theorem 1 is in Appendix A.1. The proof idea is that if the anchor features are known, then we are given noisy transformations of the coordinates of $\boldsymbol{z}$. Knowing these noisy transforms allows us to identify $\boldsymbol{z}$ up to coordinate-wise transform.

**Unknown anchor features.** The second result proves that the anchor features can be determined from an approximation of the covariance matrix of the data $\boldsymbol{X}$.

This result requires additional assumptions. The next assumption concerns the weights of the neural network $f_\theta$. This assumption needs some additional notation. After applying the chain rule to the first layer, we rewrite the derivative of mapping $f_\theta$ as follows:

$$\frac{\partial (f_\theta(\boldsymbol{w}_j \odot \boldsymbol{z}_i))_j}{\partial z_{ik}} = \sum_{d=1}^{D_1} \frac{\partial m_\theta(\boldsymbol{u}_{ij\cdot})_j}{\partial u_{ijd}} H_{dk}^{(1)} w_{jk}, \tag{15}$$

where $\{H_{dk}^{(1)}\}_{d,k=1}^{D_1,K}$ are the weights of the first neural network layer and $D_1$ is the dimension of the first layer, $u_{ijd} = \sum_{k=1}^{K} H_{dk}^{(1)} w_{jk} z_{ik}$ is the first layer before the activation function, and $m_\theta : \mathbb{R}^{D_1} \to \mathbb{R}^G$ is the rest of the neural network which takes as input the first layer $\boldsymbol{u}_{ij\cdot} = (u_{ijd})_{d=1}^{D_1}$. Let $B_{ijk} = \sum_{d=1}^{D_1} \frac{\partial m_\theta(\boldsymbol{u}_{ij\cdot})_j}{\partial u_{ijd}} H_{dk}^{(1)}$.

**A2.** Suppose $j$ is an anchor feature for factor $k$. For another feature $l$, we assume that

$$|w_{jk}| \sum_{i=1}^{N} |B_{ijk}|^2 \geq \sum_{i=1}^{N} \sum_{k'=1}^{K} |w_{lk'}||B_{ijk}||B_{ilk'}|, \tag{16}$$

with equality when $l$ is also an anchor feature for $k$ and inequality otherwise.

This assumption ensures that the covariance between two anchor features is larger than the covariance between an anchor feature and a non-anchor feature. This consequence then allows us to pinpoint the

anchor features in the covariance matrix of the data. In Appendix A.3, we show this assumption holds for a neural network with ReLU activations and independent weights.

Finally, we also require an assumption regarding the covariance of the factors: this is the same as assumption (iii) of Bing et al. (2020), which studies linear models.

**A3.** Denote the true covariance matrix of the factors as $\text{Cov}(\boldsymbol{z}_i) = C$. We assume $\min\{C_{kk}, C_{k'k'}\} > |C_{kk'}|$, for all $k, k' = 1, \ldots, K$.

Assumption A3 requires that the variance of each factor be greater than its covariance with any other factor. To gain some intuition, consider a dataset of news articles where the latent factors are topics, sports and politics. Assumption A3 would be violated if every article about sports also discussed politics (and vice versa). In this case, the anchor word for sports will be equally as correlated with an anchor word for politics as it is with another anchor word for sports. That is, there is no discernible difference in the data that helps us distinguish the sports and politics factors.

**Theorem 2.** *Assume the model Eq. 10 with $A1-3$ holds. Then, the set of anchor features can be determined uniquely from $\frac{1}{N}\sum_{i=1}^{N} Cov(\boldsymbol{x}_i)$ as $N \to \infty$ (given additional regularity conditions detailed in Appendix A.2).*

The proof of Theorem 2 is in Appendix A.2.

**Proof idea:** We adapt the proof technique of Bing et al. (2020). The proof idea of that paper is that for any two features $\boldsymbol{x}_{\cdot j}$ and $\boldsymbol{x}_{\cdot j'}$ that are anchors for the same factor, their covariance $\text{Cov}(\boldsymbol{x}_{\cdot j}, \boldsymbol{x}_{\cdot j'})$ will be greater than their covariance with any other non-anchor feature (under some conditions on the loadings matrix, analogous to A2). Then, the anchor features can be pinpointed from the observed covariance matrix. In the nonlinear case, we can apply a similar strategy: even if the mapping is nonlinear, if the mapping is the same for the anchor features, we can pinpoint the two anchors in the covariance matrix.

**Connection to the sparse VAE algorithm.** Theorem 2 proves that the existence of anchor features implies a particular structure in the covariance of the data. Consequently, an appropriate estimation procedure can then pinpoint the anchor features. In the sparse VAE, we do not directly consider the covariance matrix as it would involve laborious hyperparameter tuning to determine which covariance entries are "close enough" to be anchor features. Instead, we estimate $\boldsymbol{W}$ with an SSL prior, motivated by the sparsity in the decoder exhibited by anchor features. That is, it is the sparse decoding for the anchor features that is needed for Theorem 2, not the SSL prior; the SSL prior is a modeling choice that will likely yield the kind of sparse mapping that satisfies the anchor assumption. The SSL prior is not the only sparsity-inducing prior that we could have used; however, we found it to work well empirically. We compare the SSL prior to the horseshoe prior (Carvalho et al., 2009) in Appendix C.4.1.

**Consistency.** An important implication of identifiability is consistency: if we learn the optimal parameters of the sparse VAE, then we will recover the true factors in the limit of infinite data. Specifically, if (i) the variational family $q_\phi(\boldsymbol{z}|\boldsymbol{x})$ contains the true posterior $p_\theta(\boldsymbol{z}|\boldsymbol{x})$ and (ii) the ELBO is maximized with respect to the parameters $\theta$, $\phi$ and $\boldsymbol{W}$, then in the limit of infinite data, the sparse VAE will learn the true factors up to coordinate-wise transform and permutation.

## 4 Experiments

We empirically study the sparse VAE using a mix of synthetic, semi-synthetic and real data[1]. We consider the following questions: 1) How well does the sparse VAE recover ground truth factors when (i) the factors are uncorrelated and (ii) the factors are correlated? 2) How does the heldout predictive performance of the sparse VAE compare to related methods? 3) How does the sparse VAE perform compared to related methods when the correlation between factors is different in the training and test data? 4) Does the sparse VAE find meaningful structure in data?

We compare the sparse VAE to non-negative matrix factorization (NMF) and algorithms for DGMs: the VAE (Kingma and Welling, 2014); $\beta$-VAE (Higgins et al., 2017); VSC (Tonolini et al., 2020); and OI-VAE

---

[1]The sparse VAE implementation may be found at `https://github.com/gemoran/sparse-vae-code`.

**(a)** Heldout mean squared error (lower is better)

**(b)** Ground truth factor recovery (higher is better)

**Figure 3:** Synthetic data. (a) Sparse VAE has better heldout predictive performance than the VAE over a range of factor correlation levels. (b) Sparse VAE recovers the true factors better than the VAE. ($\beta$-VAE performed similarly to VAE). Scores are shown for 25 datasets per correlation setting.

(Ainsworth et al., 2018). None of the underlying DGMs have identifiability guarantees. We find that: 1) on synthetic data, the sparse VAE achieves the best recovery of ground truth factors both when the factors are correlated and uncorrelated; 2) the sparse VAE achieves the best predictive performance on both synthetic and real datasets; 3) the sparse VAE achieves the best predictive performance on heldout data that is generated from factors that different correlation to those in the training data; and (4) the sparse VAE finds interpretable factors in both text and movie-ratings data. On a single-cell genomics dataset, the sparse VAE learns factors which predict cell type with high accuracy.

**Datasets.** We consider the following datasets; further details about the data are given in Appendix C.1.

- **Synthetic data.** We consider again the synthetic dataset generated as Eq. 9. We vary correlation between the true factors from $\rho = 0, 0.2, 0.4, 0.6, 0.8$.

- **PeerRead** (Kang et al., 2018). Dataset of word counts for paper abstracts ($N \approx 10,000, G = 500$).

- **MovieLens** (Harper and Konstan, 2015). Dataset of binary user-movie ratings ($N = 100,000, G = 300$).

- **Semi-synthetic PeerRead.** A semi-synthetic version of the PeerRead dataset in which the correlations between data-generating factors differ across the test and training data, inducing different training and test distributions.

- **Zeisel** (Zeisel et al., 2015). Dataset of RNA molecule counts in mouse cortex cells ($N = 3005, G = 558$).

**Implementation details.** Empirically, we found that setting the prior covariance of the factors to $\Sigma_z = I_K$ worked well, even when the true generative factors were correlated (Figure 3a). Further implementation details for all methods are in Appendix C.

**Recovering the ground truth factors.** How well does the sparse VAE recover the factors when the ground truth factors are known? We assess this question with simulated data. We use the DCI disentanglement score to measure the fidelity between estimated factors and true factors (Eastwood and Williams, 2018). The DCI score is an average measure of how relevant each estimated factor is for the true factors. This score also penalizes estimated factors that are equally informative for multiple true factors.

We create synthetic datasets with the model given in Eq. 9 and evaluate the DCI metric between estimated and true factors as the true factors are increasingly correlated. As the correlation increases, we expect a standard VAE to conflate the factors while the sparse VAE recovers the two true underlying factors.

We see this phenomenon in Figure 3b; the sparse VAE has higher DCI scores than the VAE in all settings. The sparse VAE is robust to factor correlations up to $\rho = 0.4$, with decreasing performance as $\rho$ is further increased. Here, VSC performs worse than both the sparse VAE and VAE (the MSE scores for VSC in were too large for visualization in Figure 3a). The poor performance of VSC is likely due to the true generative factors in Eq. 9 not being sparse; only the mapping between the factors and features is sparse. Note that we do not compare with NMF in this example; as NMF is a linear method, it cannot reconstruct the nonlinear data generation process using only two latent dimensions.

**Heldout reconstruction.** We compare the negative log-likelihood on heldout data achieved by the sparse VAE and related methods. All results are averaged over five splits of the data, with standard deviation in

**Table 1:** On movie-ratings and text data, the sparse VAE achieves the lowest heldout negative log-likelihood (NLL). For the $\beta$-VAE, we show the result with best performing $\beta$.

**(a)** MovieLens

| Method | NLL | Recall@5 | NDCG@10 |
|---|---|---|---|
| Sparse VAE | **170.9 (2.1)** | **0.98 (0.002)** | **0.98 (0.003)** |
| VAE | 175.9 (2.4) | 0.97 (0.001) | 0.96 (0.001) |
| $\beta$-VAE ($\beta = 2$) | 178.2 (2.4) | 0.95 (0.002) | 0.93 (0.002) |
| OI-VAE | 212.1 (2.6) | 0.51 (0.004) | 0.50 (0.004) |
| VSC | 192.2 (2.3) | 0.79 (0.008) | 0.77 (0.009) |
| NMF | 198.6 (2.5) | 0.90 (0.004) | 0.85 (0.004) |

**(b)** PeerRead

| Method | NLL |
|---|---|
| Sparse VAE | **245.0 (2.0)** |
| VAE | 252.6 (1.4) |
| $\beta$-VAE ($\beta = 2$) | 254.5 (3.0) |
| OI-VAE | 260.6 (1.2) |
| VSC | 252.9 (2.0) |
| NMF | 267.2 (1.0) |

**Table 2:** On the semi-synthetic PeerRead data, the sparse VAE achieves the lowest heldout negative log-likelihood. We create three settings where the difference between training and test data distributions range from high (hardest) to low (easiest).

| | Difference between train and test | | |
|---|---|---|---|
| | High | Medium | Low |
| Method | Negative log-likelihood | | |
| Sparse VAE | **52.4 (0.4)** | **49.2 (0.3)** | **48.6 (0.1)** |
| VAE | 54.6 (0.5) | 52.3 (0.2) | 50.8 (0.2) |
| $\beta$-VAE ($\beta = 2$) | 54.7 (0.3) | 52.1 (0.2) | 51 (0.4) |
| OI-VAE | 65.7 (0.3) | 64.4 (0.3) | 64.6 (0.2) |
| VSC | 58.7 (0.6) | 56.1 (0.3) | 55.4 (0.2) |
| NMF | 60.1 (0.02) | 58.0 (0.08) | 57.3 (0.6) |

parentheses. Figure 3a shows that the sparse VAE performs better than the compared methods on synthetic data. On MovieLens and PeerRead datasets, Tables 1a and 1b show that the SparseVAE achieves the lowest heldout negative log-likelihood among the compared methods. For the MovieLens dataset, Table 1a additionally shows that the sparse VAE has the highest heldout Recall@5 and normalized discounted cumulative gain (NDCG@10), which compare the predicted rank of heldout items to their true rank (Liang et al., 2018).

**Different training and test distributions.** How does the sparse VAE perform when the factors that generate data are correlated differently across training and test distributions? This particular type of shift in distribution affects many real world settings. For example, we may estimate document representations from scientific papers where articles about machine learning also often discuss genomics, and want to use the representations to analyze new collections of papers where articles about machine learning rarely involve genomics. We hypothesize that because the sparse VAE associates each latent factor with only a few features (e.g., words) even when the factors are correlated, it will reconstruct differently distributed test data better than the related methods.

We assess this question using the semi-synthetic PeerRead dataset, where the train and test data were generated by factors with different correlations. Table 2 summarizes the results from three settings where the difference between training and test data distributions range from high (hardest) to low (easiest). We report the average negative log-likelihood across five semi-simulated datasets. The sparse VAE performs the best, highlighting its ability to estimate models which generalize better to test data where the factors have a different distribution.

**Interpretability.** We now examine whether the sparse VAE finds interpretable structure in the data. For each factor in the MovieLens dataset, we consider the four movies with the largest selector variable $\widehat{w}_{jk}$ values (Table 3a). The sparse VAE finds clear patterns: for example, the top movies in first factor are all science fiction movies; the second factor contains animated children's movies; the third factor contains three Star Wars movies and an Indiana Jones movie, all blockbuster science fiction movies from the 1980s.

**Table 3:** On movie-ratings and text data, the sparse VAE finds meaningful topics via the matrix $\boldsymbol{W}$.

<table>
<tr><td colspan="2" align="center">(a) MovieLens</td><td colspan="2" align="center">(b) PeerRead</td></tr>
<tr><td>Topic</td><td>Movies</td><td>Topic</td><td>Words</td></tr>
<tr><td>A</td><td>The Fifth Element; Alien; Gattaca; Aliens</td><td>A</td><td>task; policy; planning; heuristic; decision</td></tr>
<tr><td>B</td><td>A Bug's Life; Monsters, Inc.; Toy Story 3 & 2</td><td>B</td><td>information; feature; complex; sparse; probability</td></tr>
<tr><td>C</td><td>Star Wars V, IV & IV; Indiana Jones 1</td><td>C</td><td>given; network; method; bayesian; neural</td></tr>
</table>

**Table 4:** On the Zeisel data, the sparse VAE has the best heldout cell type prediction accuracy, followed closely by a VAE.

| Method | Sparse VAE | VAE ($\beta = 1$) | NMF | VSC | OIVAE |
|---|---|---|---|---|---|
| Accuracy | **0.95 (0.003)** | 0.94 (0.016) | 0.91 (0.007) | 0.89 (0.062) | 0.80 (0.019) |

We perform the same analysis for the PeerRead dataset (Table 3b). The sparse VAE finds some themes of computer science papers: the first factor contains words related to reinforcement learning; the second factor contains words about information theory; the third factor involves words about neural networks. We note that NMF also finds interpretable topics in both MovieLens and PeerRead (Appendix C.5.1); however, NMF has worse heldout predictive performance than the sparse VAE (Tables 1a and 1b). The sparse VAE retains the interpretability of NMF while incorporating flexible function estimation.

Finally, we consider the single-cell genomics dataset of Zeisel et al. (2015). We examine how well the factors learned by each method predict the cell type in a multinomial regression (note the cell type is not used when learning the factors). The sparse VAE is the best performing method (Table 4). Although the VAE is competitive in this setting, we see that the sparse VAE does better than methods such as OI-VAE and VSC that were designed to produce interpretable results.

## 5 Discussion

We developed the sparse DGM, a model with SSL priors that induce sparsity in the mapping between factors and features. Under an anchor feature assumption, we proved that the sparse DGM model has identifiable factors. To fit the sparse DGM, we develop the sparse VAE algorithm. On real and synthetic data, we show the sparse VAE performs well. (i) It has good heldout predictive performance. (ii) It generalizes to out of distribution data. (iii) It is interpretable.

There are a few limitations of this work. First, the sparse DGM is designed for tabular data, where each feature $j$ has a consistent meaning across samples. Image data does not have this property as a specific pixel has no consistent meaning across samples. Second, the sparse VAE algorithm is more computationally intensive than a standard VAE. This expense is because at each gradient step, it requires a forward pass through the network for each of the $G$ features. Finally, we did not provide theoretical estimation guarantees for the sparse VAE algorithm. Similarly to other algorithms for fitting DGMs, estimation guarantees for the sparse VAE are difficult to obtain due to the nonconvexity of the objective function. The majority of works which consider identifiability in DGMs also do not provide estimation guarantees for their algorithms, including Locatello et al. (2020); Khemakhem et al. (2020). We leave such theoretical study of the sparse VAE algorithm to future work.

**Acknowledgments**

Funding to support this research was provided for by the Eric and Wendy Schmidt Center, the Canada-CIFAR AI Chairs program, National Science Foundation Grant NSF-CHE-2231174, NSF IIS 2127869, ONR N00014-17-1-2131, ONR N00014-15-1-2209, the Simons Foundation, the Sloan Foundation, and Open Philanthropy.

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

# A Proofs

## A.1 Proof of Theorem 1

We consider the case of known anchor features, where we have at least two anchor features per factor. We consider the case where the likelihood is Gaussian. We have two solutions, $(\widetilde{\theta}, \widetilde{z}, \widetilde{\sigma}^2)$ and $(\widehat{\theta}, \widehat{z}, \widehat{\sigma}^2)$, with equal likelihood. We show that $\widetilde{z}$ must be a coordinate-wise transform of $\widehat{z}$.

We have

$$\sum_{i=1}^{N} \sum_{j=1}^{G} \frac{1}{\widetilde{\sigma}_j^2} [x_{ij} - f_{\widetilde{\theta}_j}(\widetilde{\boldsymbol{w}}_j \odot \widetilde{\boldsymbol{z}}_i)]^2 = \sum_{i=1}^{N} \sum_{j=1}^{G} \frac{1}{\widehat{\sigma}_j^2} [x_{ij} - f_{\widehat{\theta}_j}(\widehat{\boldsymbol{w}}_j \odot \widehat{\boldsymbol{z}}_i)]^2, \tag{17}$$

where $f_{\theta_j}(\cdot)$ denotes $f_\theta(\cdot)_j$, the $j$th output of $f_\theta$.

We prove that, given the anchor feature for factor $k$, $z_{ik}$ is identifiable. As all factors are assumed to have anchor features, this holds for all $k \in \{1, \dots, K\}$. Suppose $j$ is an anchor feature for $k$. That is, $w_{jl} = 0$ for all $l \neq k$. Then, $\widetilde{\boldsymbol{w}}_j \odot \widetilde{\boldsymbol{z}}_i = (0, \dots, 0, \widetilde{w}_{jk}\widetilde{z}_{ik}, 0, \dots, 0)$. With slight abuse of notation, we then write $f_{\widetilde{\theta}_j}(\widetilde{\boldsymbol{w}}_j \odot \widetilde{\boldsymbol{z}}_i) = f_{\widetilde{\theta}_j}(\widetilde{w}_{jk}\widetilde{z}_{ik})$.

Now, for Eq. 17 to hold for any value of $\{x_{ij}\}_{i,j=1}^{N,G}$ we must have, for all $j = 1, \dots, G$:

$$\sum_{i=1}^{N} \frac{1}{\widetilde{\sigma}_j^2} [x_{ij} - f_{\widetilde{\theta}_j}(\widetilde{w}_{jk}\widetilde{z}_{ik})]^2 = \sum_{i=1}^{N} \frac{1}{\widehat{\sigma}_j^2} [x_{ij} - f_{\widehat{\theta}_j}(\widehat{w}_{jk}\widehat{z}_{ik})]^2 \tag{18}$$

$$\left( \frac{1}{\widetilde{\sigma}_j^2} - \frac{1}{\widehat{\sigma}_j^2} \right) \sum_{i=1}^{N} x_{ij}^2 - 2 \sum_{i=1}^{N} x_{ij} \left( \frac{f_{\widetilde{\theta}_j}(\widetilde{w}_{jk}\widetilde{z}_{ik})}{\widetilde{\sigma}_j^2} - \frac{f_{\widehat{\theta}_j}(\widehat{w}_{jk}\widehat{z}_{ik})}{\widehat{\sigma}_j^2} \right) + \sum_{i=1}^{N} \left( \frac{f_{\widetilde{\theta}_j}^2(\widetilde{w}_{jk}\widetilde{z}_{ik})}{\widetilde{\sigma}_j^2} - \frac{f_{\widehat{\theta}_j}^2(\widehat{w}_{jk}\widehat{z}_{ik})}{\widehat{\sigma}_j^2} \right) = 0. \tag{19}$$

For Eq. 19 to hold for all $x_{ij}$, we must have the coefficients equal to zero:

$$\frac{1}{\widetilde{\sigma}_j^2} - \frac{1}{\widehat{\sigma}_j^2} = 0 \implies \widetilde{\sigma}_j^2 = \widehat{\sigma}_j^2 \tag{20}$$

and

$$\frac{f_{\widetilde{\theta}_j}(\widetilde{w}_{jk}\widetilde{z}_{ik})}{\widetilde{\sigma}_j^2} - \frac{f_{\widehat{\theta}_j}\widehat{w}_{jk}(\widehat{z}_{ik})}{\widehat{\sigma}_j^2} = 0 \implies f_{\widetilde{\theta}_j}(\widetilde{w}_{jk}\widetilde{z}_{ik}) = f_{\widehat{\theta}_j}(\widehat{w}_{jk}\widehat{z}_{ik}). \tag{21}$$

If $f_{\widetilde{\theta}_j} : \mathbb{R} \to \mathbb{R}$ is invertible, then we have

$$\widetilde{z}_{ik} = \widetilde{w}_{jk}^{-1} f_{\widetilde{\theta}_j}^{-1}(f_{\widehat{\theta}_j}(\widehat{w}_{jk}\widehat{z}_{ik})). \tag{22}$$

That is, $z_{ik}$ is identifiable up to coordinate-wise transformation.

If $f_{\theta_j}$ is not invertible, then there exists bijective functions $g : \mathbb{R} \to \mathbb{R}$, $h : \mathbb{R} \to \mathbb{R}$ such that $\widehat{w}_{jk}\widehat{z}_{ik} = g(\widetilde{w}_{jk}\widetilde{z}_{ik})$ and the equality is satisfied

$$f_{\widetilde{\theta}_j}(\widetilde{w}_{jk}\widetilde{z}_{ik}) = f_{\widetilde{\theta}_j}(h(g(\widetilde{w}_{jk}\widetilde{z}_{ik}))) \quad \text{(as } f_{\theta_j} \text{ is not invertible, } \widetilde{w}_{jk}\widetilde{z}_{ik} \neq h(g(\widetilde{w}_{jk}\widetilde{z}_{ik}))\text{)} \tag{23}$$

$$= f_{\widehat{\theta}_j}(\widehat{w}_{jk}\widehat{z}_{ik}). \tag{24}$$

As we have $\widehat{z}_{ik} = \widehat{w}_{jk}^{-1} g(\widetilde{w}_{jk}\widetilde{z}_{ik})$, $z_{ik}$ is identifiable up to coordinate-wise transformation.

### A.2 Proof of Theorem 2

To prove the result, we:

1. Approximate the covariance $\frac{1}{N}\sum_{i=1}^{N}\mathrm{Cov}(\boldsymbol{x}_i)$ using a first order Taylor approximation.

2. Show that an anchor feature for factor $k$ always has larger covariance with another anchor feature for $k$ than other non-anchor features. Specifically: for any $k \in \{1,\ldots,K\}$ and $j =$ anchor for $k$, we have

   - $\sum_{i=1}^{N}\mathrm{Cov}(x_{ij},x_{il}) = C_{kk}w_{jk}^2\sum_{i=1}^{N}B_{ijk}^2$ for all $l =$ anchor for $k$,
   - $\sum_{i=1}^{N}\mathrm{Cov}(x_{ij},x_{il}) < C_{kk}w_{jk}^2\sum_{i=1}^{N}B_{ijk}^2$ for all $l \neq$ anchor for $k$.

3. Apply Theorem 1 of Bing et al. (2020) to prove that the anchor features can be identified from the covariance of the data.

**Regularity condition.** Let $\mu(\boldsymbol{z}_i) = \mathbb{E}[\boldsymbol{x}_i|\boldsymbol{z}_i]$. We assume: for all $\boldsymbol{z}_i$ in a neighborhood of $\widehat{\boldsymbol{z}}_i = \mathbb{E}[\boldsymbol{z}_i]$,

$$\frac{1}{N}\sum_{i=1}^{N}\mu(\boldsymbol{z}_i) = \frac{1}{N}\sum_{i=1}^{N}\widehat{\boldsymbol{z}}_i + \frac{\partial\mu(\boldsymbol{z}_i)}{\partial\boldsymbol{z}_i}\Big|_{\boldsymbol{z}_i=\widehat{\boldsymbol{z}}_i}(\boldsymbol{z}_i - \widehat{\boldsymbol{z}}_i) + o_P(1) \tag{25}$$

where $o_P(1)$ denotes a random variable converging to 0 in probability.

**Step 1.** Here is the marginal covariance of the Sparse VAE:

$$\mathrm{Var}(\boldsymbol{x}_i) = \mathbb{E}\left[\mathrm{Var}(\boldsymbol{x}_i|\boldsymbol{z}_i)\right] + \mathrm{Var}(\mathbb{E}\left[\boldsymbol{x}_i|\boldsymbol{z}_i\right]) \tag{26}$$
$$= \boldsymbol{\Sigma} + \mathrm{Var}(\mu(\boldsymbol{z}_i)) \tag{27}$$

Take the first order Taylor expansion $\mu(\boldsymbol{z}_i) \approx \widehat{\boldsymbol{z}}_i + \frac{\partial\mu(\boldsymbol{z}_i)}{\partial\boldsymbol{z}_i}\big|_{\boldsymbol{z}_i=\widehat{\boldsymbol{z}}_i}(\boldsymbol{z}_i - \widehat{\boldsymbol{z}}_i)$.

Then

$$\mathrm{Var}(\mu(\boldsymbol{z}_i)) \approx \frac{\partial\mu(\boldsymbol{z}_i)}{\partial\boldsymbol{z}_i}\Big|_{\boldsymbol{z}_i=\widehat{\boldsymbol{z}}_i}\mathrm{Var}(\boldsymbol{z}_i)\left(\frac{\partial\mu(\boldsymbol{z}_i)}{\partial\boldsymbol{z}_i}\Big|_{\boldsymbol{z}_i=\widehat{\boldsymbol{z}}_i}\right)^T. \tag{28}$$

Now, $\mu(\boldsymbol{z}_i) = \{f_{\theta_j}(\boldsymbol{w}_j \odot \boldsymbol{z}_i)\}_{j=1}^{G}$. Then, for any two features, $j$ and $l$, we have

$$\mathrm{Cov}(x_{ij},x_{il}) \approx \sum_{k=1}^{K}\sum_{k'=1}^{K}\frac{\partial f_{\theta_j}(\boldsymbol{w}_j \odot \boldsymbol{z}_i)}{\partial z_{ik}}\Big|_{\boldsymbol{z}_i=\widehat{\boldsymbol{z}}_i}\frac{\partial f_{\theta_l}(\boldsymbol{w}_l \odot \boldsymbol{z}_i)}{\partial z_{ik'}}\Big|_{\boldsymbol{z}_i=\widehat{\boldsymbol{z}}_i}C_{kk'} \tag{29}$$

where $\mathrm{Var}(\boldsymbol{z}_i) = C$.

**Step 2.** We re-write $f_{\theta_j}(\cdot)$ to expose the first layer of the neural network (before the activation function is applied):

$$f_{\theta_j}(\boldsymbol{w}_j \odot \boldsymbol{z}_i) = m_{\theta_j}(\boldsymbol{u}_{ij\cdot}) \tag{30}$$

where $\boldsymbol{u}_{ij\cdot} = \{u_{ijd}\}_{d=1}^{D_1}$ with

$$u_{ijd} = \sum_{k=1}^{K}H_{dk}^{(1)}w_{jk}z_{ik}, \tag{31}$$

where $\{H_{dk}^{(1)}\}_{d,k=1}^{D_1,K}$ are the weights of the first neural network layer with dimension $D_1$.

Then,

$$\text{Cov}(x_{ij}, x_{il}) = \sum_{k=1}^{K} \sum_{k'=1}^{K} w_{jk} w_{lk'} \left( \sum_{d=1}^{D_1} H_{dk}^{(1)} \frac{\partial m_{\theta_j}(\boldsymbol{u}_{ij \cdot})}{\partial u_{ijd}} \right) \left( \sum_{d=1}^{D_1} H_{dk'}^{(1)} \frac{\partial m_{\theta_l}(\boldsymbol{u}_{il \cdot})}{\partial u_{ild}} \right) C_{kk'} \tag{32}$$

$$= \sum_{k=1}^{K} \sum_{k'=1}^{K} w_{jk} w_{lk'} B_{ijk} B_{ilk'} C_{kk'}, \tag{33}$$

where $B_{ijk} = \sum_{d=1}^{D_1} H_{dk}^{(1)} \frac{\partial m_{\theta_j}(\boldsymbol{u}_{ij \cdot})}{\partial u_{ijd}}$.

Suppose $j$ is an anchor feature for $k$. Then the absolute covariance of feature $j$ and any other feature $l$ is:

$$|\text{Cov}(x_{ij}, x_{il})| = |w_{jk} B_{ijk} \sum_{k'=1}^{K} w_{lk'} B_{ilk'} C_{kk'}| \tag{34}$$

$$\leq |C_{kk} w_{jk} B_{ijk} \sum_{k'=1}^{K} w_{lk'} B_{ilk'}| \quad \text{by A3.} \tag{35}$$

$$\leq C_{kk} w_{jk}^2 B_{ijk}^2 \quad \text{by A2.} \tag{36}$$

**Step 3.** In Step 2, we proved the equivalent of Lemma 2 of Bing et al. (2020), adapted for the Sparse VAE. This allows us to apply Theorem 1 of Bing et al. (2020), which proves that the anchor features can be determined uniquely from the approximate covariance matrix, $\frac{1}{N} \sum_{i=1}^{N} \text{Cov}(\boldsymbol{x}_i)$, as $N \to \infty$.

### A.3 Discussion of identifiability assumptions

In this section, we examine the suitability of assumption A2. We do so by showing A2 holds for a three-layer neural network with ReLU activation functions, independently distributed Gaussian weights and no bias terms. Specifically:

$$\mathbb{E}[x_{ij}|\boldsymbol{w}_j, \boldsymbol{z}_i] = \boldsymbol{H}_{j \cdot}^{(3)} \text{ ReLU}(\boldsymbol{H}^{(2)} \text{ ReLU}(\boldsymbol{H}^{(1)}(\boldsymbol{w}_j \odot \boldsymbol{z}_i))) \tag{37}$$

where $\boldsymbol{H}^{(1)} \in \mathbb{R}^{D \times K}$ are the weights for layer 1 with $D$ hidden units, $\boldsymbol{H}^{(2)} \in \mathbb{R}^{D \times D}$ are the weights for layer 2, and $\boldsymbol{H}_{j \cdot}^{(3)}$ denotes the $j$th row of the weights for layer 3, $\boldsymbol{H}^{(3)} \in \mathbb{R}^{G \times D}$.

We have

$$\frac{\partial \mu(\boldsymbol{z}_i)_j}{\partial z_{ik}} = \sum_{d_1=1}^{D} \sum_{d_2=1}^{D} H_{j,d_2}^{(3)} H_{d_2,d_1}^{(2)} H_{d_1,k}^{(1)} w_{jk} \mathbb{I}\left[\boldsymbol{H}_{d_2 \cdot}^{(2)} \text{ReLU}(\boldsymbol{H}^{(1)}(\boldsymbol{w}_j \odot \boldsymbol{z}_i)) > 0\right] \mathbb{I}\left[\sum_{k=1}^{K} H_{d_1,k}^{(1)} w_{jk} z_{ik} > 0\right]. \tag{38}$$

where $\mathbb{I}(\cdot)$ is the indicator function.

Assume $\{w_{jk}\}_{j,k=1}^{G,K}$ are independent and symmetric around zero. Assume all weights are independent and distributed as: $H_{d_1,d_2}^{(m)} \overset{i.i.d.}{\sim} N(0, \tau)$ for all layers $m = 1, 2, 3$.

Taking the first order Taylor approximation, for any two features, $j$ and $l$, we have

$$\text{Cov}(x_{ij}, x_{il}) \approx \sum_{k=1}^{K} \sum_{k'=1}^{K} C_{kk'} \sum_{d_1=1}^{D} \sum_{d_2=1}^{D} \sum_{d_1'=1}^{D} \sum_{d_2'=1}^{D} (H_{j,d_2}^{(3)} H_{d_2,d_1}^{(2)} H_{d_1,k}^{(1)} w_{jk})(H_{l,d_2'}^{(3)} H_{d_2',d_1'}^{(2)} H_{d_1',k'}^{(1)} w_{lk'}) \tag{39}$$

$$\times \mathbb{I}\left[\boldsymbol{H}_{d_2 \cdot}^{(2)} \text{ReLU}(\boldsymbol{H}^{(1)}(\boldsymbol{w}_j \odot \widehat{\boldsymbol{z}}_i)) > 0\right] \mathbb{I}\left[\sum_{k=1}^{K} H_{d_1,k}^{(1)} w_{jk} \widehat{z}_{ik} > 0\right] \tag{40}$$

$$\times \mathbb{I}\left[\boldsymbol{H}_{d_2' \cdot}^{(2)} \text{ReLU}(\boldsymbol{H}^{(1)} \boldsymbol{w}_l \odot \widehat{\boldsymbol{z}}_i) > 0\right] \mathbb{I}\left[\sum_{k=1}^{K} H_{d_1',k}^{(1)} w_{lk} \widehat{z}_{ik} > 0\right]. \tag{41}$$

As the neural network weights are independent Gaussians, we keep only the terms where $d_1 = d_1'$ and $d_2 = d_2'$:

$$\text{Cov}(x_{ij}, x_{il}) \approx \sum_{d_2=1}^{D} H_{j,d_2}^{(3)} H_{l,d_2}^{(3)} \sum_{d_1=1}^{D} (H_{d_2,d_1}^{(2)})^2 \sum_{k=1}^{K} \sum_{k'=1}^{K} C_{kk'} H_{d_1,k}^{(1)} H_{d_1,k'}^{(1)} w_{jk} w_{lk'} \tag{42}$$

$$\times \mathbb{I}\left[ \boldsymbol{H}_{d_2}^{(2)} \text{ReLU}(\boldsymbol{H}^{(1)}(\boldsymbol{w}_j \odot \widehat{\boldsymbol{z}}_i)) > 0 \right] \mathbb{I}\left[ \sum_{k=1}^{K} H_{d_1,k}^{(1)} w_{jk} \widehat{z}_{ik} > 0 \right] \tag{43}$$

$$\times \mathbb{I}\left[ \boldsymbol{H}_{d_2}^{(2)} \text{ReLU}(\boldsymbol{H}^{(1)} \boldsymbol{w}_l \odot \widehat{\boldsymbol{z}}_i) > 0 \right] \mathbb{I}\left[ \sum_{k=1}^{K} H_{d_1,k}^{(1)} w_{lk} \widehat{z}_{ik} > 0 \right]. \tag{44}$$

If $j$ and $l$ are both anchor features for factor $k$, we have:

$$\text{Cov}(x_{ij}, x_{il}) \approx \sum_{d_2=1}^{D} (H_{j,d_2}^{(3)})^2 \sum_{d_1=1}^{D} (H_{d_2,d_1}^{(2)})^2 C_{kk} (H_{d_1,k}^{(1)})^2 w_{jk}^2 \tag{45}$$

$$\times \mathbb{I}\left[ \boldsymbol{H}_{d_2}^{(2)} \text{ReLU}(\boldsymbol{H}^{(1)}(\boldsymbol{w}_j \odot \widehat{\boldsymbol{z}}_i)) > 0 \right] \mathbb{I}\left[ \sum_{k=1}^{K} H_{d_1,k}^{(1)} w_{jk} \widehat{z}_{ik} > 0 \right], \tag{46}$$

as $H_{j,d_2}^{(3)} = H_{l,d_2}^{(3)}$ and $w_{jk} = w_{lk}$ by definition of an anchor feature.

Meanwhile, if $j$ and $l$ are not anchor features for the same factor, we have

$$\text{Cov}(x_{ij}, x_{il}) \approx \sum_{d_2=1}^{D} H_{j,d_2}^{(3)} H_{l,d_2}^{(3)} A_{d_2} \tag{47}$$

where

$$A_{d_2} = \sum_{d_1=1}^{D} (H_{d_2,d_1}^{(2)})^2 \sum_{k=1}^{K} \sum_{k'=1}^{K} C_{kk'} H_{d_1,k}^{(1)} H_{d_1,k'}^{(1)} w_{jk} w_{lk'} \mathbb{I}\left[ \boldsymbol{H}_{d_2}^{(2)} \text{ReLU}(\boldsymbol{H}^{(1)}(\boldsymbol{w}_j \odot \widehat{\boldsymbol{z}}_i)) > 0 \right] \tag{48}$$

$$\times \mathbb{I}\left[ \sum_{k=1}^{K} H_{d_1,k}^{(1)} w_{jk} \widehat{z}_{ik} > 0 \right] \mathbb{I}\left[ \boldsymbol{H}_{d_2}^{(2)} \text{ReLU}(\boldsymbol{H}^{(1)} \boldsymbol{w}_l \odot \widehat{\boldsymbol{z}}_i) > 0 \right] \mathbb{I}\left[ \sum_{k=1}^{K} H_{d_1,k}^{(1)} w_{lk} \widehat{z}_{ik} > 0 \right]. \tag{49}$$

As the weights are independent, we have that $H_{j,d_2}^{(3)}$, $H_{l,d_2}^{(3)}$, $A_{d_2}$ are independent. For large $D$, we then have

$$\sum_{d_2=1}^{D} H_{j,d_2}^{(3)} H_{l,d_2}^{(3)} A_{d_2} \to 0. \tag{50}$$

Hence, for two anchor features $j$ and $l$ and non-anchor feature $m$, we have $\text{Cov}(x_{ij}, x_{il}) > \text{Cov}(x_{ij}, x_{im})$.

# B    Inference

## B.1    Derivation of ELBO for identity prior covariance

In the main body of the paper, we set the prior covariance to $\boldsymbol{\Sigma}_z = \boldsymbol{I}$ as it performed well empirically and was computationally efficient. For this setting, the objective is derived as

$$\sum_{i=1}^{N} \log p_\theta(\boldsymbol{x}_i, \boldsymbol{W}, \boldsymbol{\eta}) = \sum_{i=1}^{N} \log \left( \int p_\theta(\boldsymbol{x}_i|\boldsymbol{z}_i, \boldsymbol{W}) p(\boldsymbol{z}_i) d\boldsymbol{z}_i \right) + \log \left( \int p(\boldsymbol{W}|\boldsymbol{\Gamma}) p(\boldsymbol{\Gamma}|\boldsymbol{\eta}) p(\boldsymbol{\eta}) d\boldsymbol{\Gamma} \right) \tag{51}$$

$$= \sum_{i=1}^{N} \log \left( \mathbb{E}_{q_\phi(\boldsymbol{z}_i|\boldsymbol{x}_i)} \left[ p_\theta(\boldsymbol{x}_i|\boldsymbol{z}_i, \boldsymbol{W}) \frac{p(\boldsymbol{z}_i)}{q_\phi(\boldsymbol{z}_i|\boldsymbol{x}_i)} \right] \right) + \log \left( \mathbb{E}_{\boldsymbol{\Gamma}|\boldsymbol{W}, \boldsymbol{\eta}} \left[ \frac{p(\boldsymbol{W}|\boldsymbol{\Gamma}) p(\boldsymbol{\Gamma}|\boldsymbol{\eta}) p(\boldsymbol{\eta})}{p(\boldsymbol{\Gamma}|\boldsymbol{W}, \boldsymbol{\eta})} \right] \right) \tag{52}$$

$$\geq \sum_{i=1}^{N} \left\{ \mathbb{E}_{q_\phi(\boldsymbol{z}_i|\boldsymbol{x}_i)}[\log p_\theta(\boldsymbol{x}_i|\boldsymbol{z}_i, \boldsymbol{W})] - D_{KL}(q_\phi(\boldsymbol{z}_i|\boldsymbol{x}_i)||p(\boldsymbol{z}_i)) \right\} + \mathbb{E}_{\boldsymbol{\Gamma}|\boldsymbol{W}, \boldsymbol{\eta}} \left[ \log[p(\boldsymbol{W}|\boldsymbol{\Gamma}) p(\boldsymbol{\Gamma}|\boldsymbol{\eta}) p(\boldsymbol{\eta})] \right]$$

$$\equiv \mathcal{L}(\theta, \phi, \boldsymbol{W}, \boldsymbol{\eta}). \tag{53}$$

The KL divergence between the variational posterior $q_\phi(\boldsymbol{z}_i|\boldsymbol{x}_i)$ and the prior $\boldsymbol{z}_i \sim \mathcal{N}_K(\boldsymbol{0}, \boldsymbol{I})$ is:

$$D_{KL}(q_\phi(\boldsymbol{z}_i|\boldsymbol{x}_i)||p(\boldsymbol{z}_i)) = -\frac{1}{2} \sum_{k=1}^{K} \left[ 1 + \log(\sigma_\phi^2(\boldsymbol{x}_i)) - (\mu_\phi(\boldsymbol{x}_i))^2 - \sigma_\phi^2(\boldsymbol{x}_i) \right] \tag{54}$$

The final term of the ELBO in Eq. 53 is:

$$\mathbb{E}_{\boldsymbol{\Gamma}|\boldsymbol{W}^{(t)}, \boldsymbol{\eta}^{(t)}} \left[ \log[p(\boldsymbol{W}|\boldsymbol{\Gamma}) p(\boldsymbol{\Gamma}|\boldsymbol{\eta}) p(\boldsymbol{\eta})] \right] = \sum_{k=1}^{K} \sum_{j=1}^{G} \lambda^*(w_{jk}^{(t)}, \eta_k^{(t)})|w_{jk}| + \sum_{k=1}^{K} \left[ \sum_{j=1}^{G} \mathbb{E}[\gamma_{jk}|w_{jk}^{(t)}, \eta_k^{(t)}] + a - 1 \right] \log \eta_k \tag{55}$$

$$+ \left[ G - \sum_{j=1}^{G} \mathbb{E}[\gamma_{jk}|w_{jk}^{(t)}, \eta_k^{(t)}] + b - 1 \right] \log(1 - \eta_k), \tag{56}$$

where

$$\mathbb{E}[\gamma_{jk}|w_{jk}, \eta_k] = \frac{\eta_k \psi_1(w_{jk})}{\eta_k \psi_1(w_{jk}) + (1 - \eta_k)\psi_0(w_{jk})} \tag{57}$$

$$\lambda^*(w_{jk}, \eta_k) = \lambda_1 \mathbb{E}[\gamma_{jk}|w_{jk}, \eta_k] + \lambda_0(1 - \mathbb{E}[\gamma_{jk}|w_{jk}, \eta_k]). \tag{58}$$

As described in Algorithm 1, we alternate between updating $\mathbb{E}[\gamma_{jk}|w_{jk}^{(t)}, \eta_k^{(t)}]$ and $\eta_k$, and taking gradient steps for $\theta, \phi$ and $\boldsymbol{W}$.

## B.2    Derivation of ELBO for general prior covariance

In this section, we detail the sparse VAE ELBO for general $\boldsymbol{\Sigma}_z$. We do not study this algorithm in this paper but we include the derivation here for completeness.

The change in the sparse VAE ELBO for general $\boldsymbol{\Sigma}_z$ is the KL divergence term:

$$D_{KL}(q_\phi(\boldsymbol{z}_i|\boldsymbol{x}_i)||p(\boldsymbol{z}_i)) = -\frac{1}{2} \left\{ \sum_{k=1}^{K}[1 + \log(\sigma_\phi^2(\boldsymbol{x}_i))] - \mathrm{tr}(\boldsymbol{\Sigma}_z^{-1}\mathrm{diag}(\sigma_\phi^2(\boldsymbol{z}_i))) - \mu_\phi(\boldsymbol{z}_i)^T \boldsymbol{\Sigma}_z^{-1} \mu_\phi(\boldsymbol{z}_i) - \log|\boldsymbol{\Sigma}_z| \right\}. \tag{59}$$

## C Details of empirical studies

### C.1 Dataset details and preprocessing

**PeerRead** (Kang et al., 2018). We discard any word tokens that appear in fewer than about 0.1% of the abstracts and in more than 90% of the abstracts. From the remaining word tokens, we consider the $G = 500$ most used tokens as features. The observations are counts of each feature across $N \approx 11,000$ abstracts.

**Semi-synthetic PeerRead** The training and testing data are distributed differently because we vary the correlations among the latent factors that generate the features across the two datasets. This data was generated as follows: (i) we applied Latent Dirichlet Allocation (LDA, Blei et al., 2003) to the PeerRead dataset using $K = 20$ components to obtain factors $\theta \in \mathbb{R}^{N \times K}$ and topics (loadings) $\beta \in \mathbb{R}^{K \times G}$. (ii) We created train set factors, $\theta_{tr}$, by dropping the last $\frac{K}{2}$ columns of $\theta$ and replacing them with columns calculated as: logit $\widetilde{\theta_{\cdot k}} = $ logit $\theta_{\cdot k} + N(0, \sigma^2)$ for each of the first $\frac{K}{2}$ latent dimension $k$. We fix the test set factors as $\theta_{te} = \theta$.

**MovieLens** (Harper and Konstan, 2015). We consider the MovieLens 25M dataset. Following (Liang et al., 2018), we code ratings four or higher as one and the remaining ratings as zero. We retain users who have rated more than 20 movies. We keep the top $G = 300$ most rated movies. Finally, we randomly subsample $N = 100,000$ users.

**Zeisel** (Zeisel et al., 2015). We first processed the data following (Zeisel et al., 2015). Next, we normalized the gene counts using quantile normalization (Bolstad et al., 2003; Bolstad, 2018). Finally, following (Lopez et al., 2018), we retained the top $G = 558$ genes with the greatest variability over cells.

### C.2 Sparse VAE settings

- For all experiments, the Sparse VAE takes the $\eta_k$ prior hyperparameters to be $a = 1, b = G$, where $G$ is the number of observed features.

- For experiments with Gaussian loss, the prior on the error variance is:

$$\sigma_j^2 \sim \text{Inverse-Gamma}(\nu/2, \nu\xi/2). \tag{60}$$

  We set $\nu = 3$. The hyperparameter $\xi$ is set to a data-dependent value. Specifically, we first calculate the sample variance of each feature, $\boldsymbol{x}_{\cdot j}$. Then, we set $\xi$ such that the 5% quantile of the sample variances is the 90% quantile of the Inverse-Gamma prior. The idea is: the sample variance is an overestimate of the true noise variance. The smaller sample variances are assumed to correspond to mostly noise and not signal – these variances are used to calibrate the prior on the noise.

### C.3 Experimental settings

- For the neural networks, we use multilayer perceptrons with the same number of nodes in each layer. These settings are detailed in Table 6.

- For stochastic optimization, we use automatic differentiation in `PyTorch`, with optimization using Adam (Kingma and Ba, 2015) with default settings (beta1=0.9, beta2=0.999)

- For LDA, we used Python's `sklearn` package with default settings. For NMF, we used `sklearn` with `alpha=1.0`.

- The dataset-specific experimental settings are in Table 6.

### C.4 Additional experiments

#### C.4.1 Synthetic data

We consider an additional experiments to answer the questions:

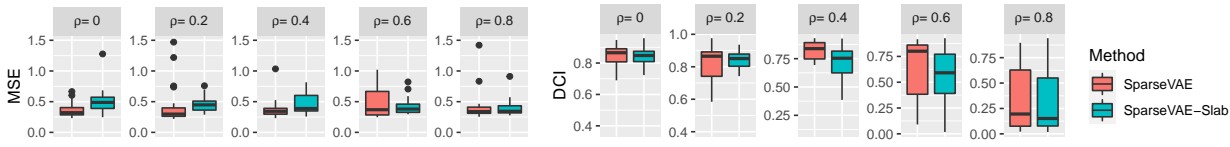

**(a)** Heldout mean squared error (lower is better)     **(b)** Ground truth factor recovery (higher is better)

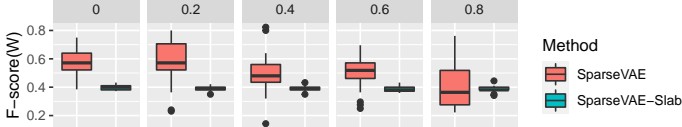

**(c)** F-score of $\boldsymbol{W}$ support recovery (higher is better)

**Figure 4:** Synthetic data. The Sparse VAE with less regularization on $\boldsymbol{W}$ (SparseVAE-Slab) performs slightly worse than Sparse VAE with more regularization on $\boldsymbol{W}$ (SparseVAE) in terms of (a) MSE; (b) DCI Disentanglement Score; and (c) F-score of the estimated support of $\boldsymbol{W}$.

1. How does the sparse VAE perform when the regularization parameter in the sparsity-inducing prior is decreased?

2. Does the Spike-and-Slab Lasso prior perform similarly to an alternate sparsity-inducing prior, the Horseshoe prior (Carvalho et al., 2009)?

**Experiment 1.** We find that the sparse VAE with less regularization ($\lambda_0 = \lambda_1 = 0.001$) performs slightly worse than the sparse VAE with $\lambda_0 = 10$, $\lambda_1 = 1$ in terms of MSE (Figure 4a) and DCI (Figure 4b). The sparse VAE with less regularization still performs well, however – this is because the $\boldsymbol{W}$ it learns also tends to be somewhat sparse (Figure 4c). Note that when $\lambda_0 = \lambda_1$, the Spike-and-Slab Lasso prior is equivalent to a Lasso prior (Tibshirani, 1996).

**Experiment 2.** We find that the Spike-and-Slab Lasso and the horseshoe perform similarly for lower values of the factor correlation $\rho$, but that the Spike-and-Slab has lower MSE and higher DCI than the horseshoe for higher values of $\rho$.

To implement the horseshoe prior, we follow Ghosh et al. (2019) and parameterize the half-Cauchy distribution $C^+(0, b)$ as:

$$a \sim C^+(0, b) \Leftrightarrow a^2 | \lambda \sim \text{Inv-Gamma}(1/2, 1/\lambda), \quad \lambda \sim \text{Inv-Gamma}(1/2, 1/b^2). \tag{61}$$

Then, we assign $\boldsymbol{W}$ a horseshoe prior:

$$\lambda_{global} \sim \text{Inv-Gamma}(1/2, 1/b_{global}^2), \tag{62}$$
$$\lambda_{local} \sim \text{Inv-Gamma}(1/2, 1/b_{local}^2), \tag{63}$$
$$\tau_{jk} | \lambda_{local} \sim \text{Inv-Gamma}(1/2, 1/\lambda_{local}), \tag{64}$$
$$v_k | \lambda_{global} \sim \text{Inv-Gamma}(1/2, 1/\lambda_{global}), \tag{65}$$
$$w_{jk} | \tau_{jk}, v_k \sim N(0, (t_{jk}^2 v_k^2) I), \tag{66}$$

where $s \sim \text{Inv-Gamma}(a, b)$ is the inverse-Gamma distribution with density $p(s) \propto s^{-a-1} \exp\{-b/s\}$ for $s > 0$. We set the hyperparameters to $b_{global} = b_{local} = 1.0$.

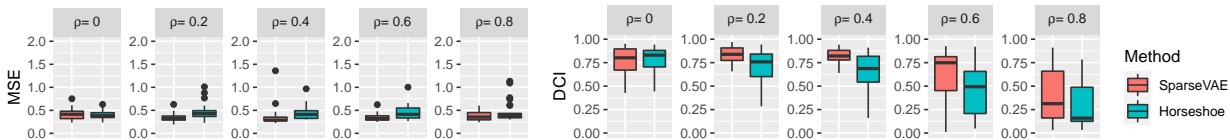

**(a)** Heldout mean squared error (lower is better)    **(b)** Ground truth factor recovery (higher is better)

**Figure 5:** Synthetic data. The Sparse VAE with the Spike-and-Slab-Lasso prior performs slightly better than Sparse VAE with a horseshoe prior $\boldsymbol{W}$ in terms of (a) MSE; (b) DCI Disentanglement Score.

With this horseshoe prior on $\boldsymbol{W}$, we again optimize $\boldsymbol{W}$ using coordinate ascent and then use closed-form updates for the inverse-Gamma parameters:

$$\tau_{jk}^{(t)} = 0.5\|w_{jk}^{(t)}\|^2/(2v_k^{(t-1)} + 1/\lambda_{local}^{(t-1)}), \tag{67}$$

$$v_k^{(t)} = 2/(G+3)\left(\sum_{j=1}^{G} w_{jk}^{(t)2}/2\tau_{jk}^2 + 1/\lambda_{global}^{(t-1)}\right), \tag{68}$$

$$\lambda_{local}^{(t)} = 2/(GK+3)\left(\sum_{k,j=1}^{K,G} 1/\tau_{jk}^{(t)2} + 1/b_{local}\right), \tag{69}$$

$$\lambda_{global}^{(t)} = 2/(K+3)\left(\sum_{k=1}^{K} 1/v_k^{(t)2} + 1/b_{global}\right). \tag{70}$$

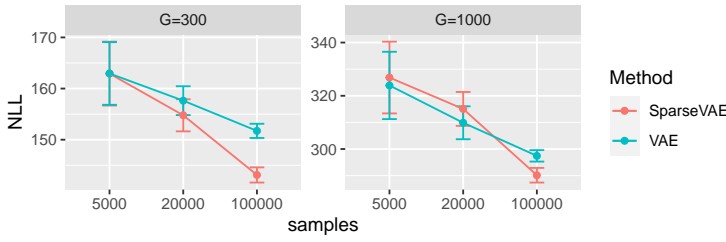

**(a)** Negative heldout log likelihood (lower is better)

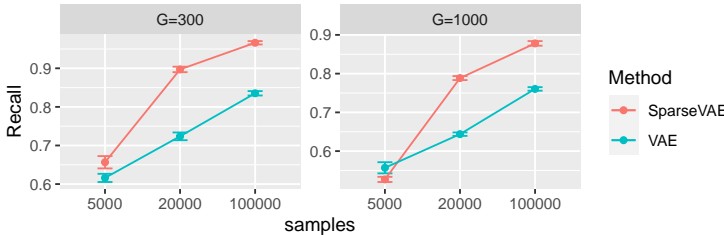

**(b)** Heldout recall (higher is better)

**Figure 6:** MovieLens. The Sparse VAE generally performs as well as or better than a VAE on (a) negative heldout log-likelihood and (b) heldout recall when varying the sample size and number of features (movies). Results are averaged over 5 splits of the data.

### C.5  MovieLens: varying sample size and number of features

In this section, we consider the performance of the Sparse VAE when we vary the sample size and number of features (movies) in the MovieLens dataset. The settings are the same as noted in Table 6, except for the latent space dimension which is set to $K = 50$.

Generally, the Sparse VAE performs as well as or better than a VAE on negative heldout log-likelihood and heldout recall (Figure 6). For the largest setting ($N = 100,000, G = 1,000$), the Sparse VAE is about 100 times slower than a VAE.

#### C.5.1  MovieLens and PeerRead: comparisons with NMF

We include additional comparisons of the Sparse VAE with non-negative matrix factorization (NMF) on the MovieLens and PeerRead datasets.

On the MovieLens data, examples of topics found by NMF include those related to science fiction; animated childrens' movies; and Star Wars (although Toy Story is included in the Star Wars topic). This performance is similar to the Sparse VAE.

On the PeerRead data, examples of topics found by NMF include those related to reinforcement learning; language modeling; and neural network models.

In summary, NMF finds interpretable topics, similarly to the Sparse VAE. However, the Sparse VAE has better heldout predictive performance than NMF. This provides evidence for the Sparse VAE retaining the interpretability of linear methods while improving model flexibility.

### C.6  Compute

- GPU: NVIDIA TITAN Xp graphics card (24GB).

- CPU: Intel E4-2620 v4 processor (64GB).

**Table 5:** On MovieLens (left) and PeerReads (right), NMF finds meaningful topics.

| Topic | Movies | Topic | Words |
|---|---|---|---|
| A | Alien; Aliens; Terminator 2; Terminator; Blade Runner | A | agent; action; policy; state; game |
| B | Shrek 1 & 2; Monsters, Inc.; Finding Nemo; The Incredibles | B | word; representation; vector; embeddings; semantic |
| C | Star Wars IV, V & VI; Toy Story | C | model; inference; neural; latent; structure |

**Table 6:** Settings for each experiment.

Synthetic data

| Settings | Value |
|---|---|
| # hidden layers | 3 |
| # layer dimension | 50 |
| Latent space dimension | 5 |
| Learning rate | 0.01 |
| Epochs | 200 |
| Batch size | 100 |
| Loss function | Gaussian |
| Sparse VAE | $\lambda_1 = 1$, $\lambda_0 = 10$ |
| $\beta$-VAE | [2, 4, 6, 8, 16] |
| VSC | $\alpha = 0.01$ |
| OI-VAE | $\lambda = 1, p = 5$ |
| Runtime per split | CPU, 2 mins |

MovieLens

| Settings | Value |
|---|---|
| # hidden layers | 3 |
| # layer dimension | 300 |
| Latent space dimension | 30 |
| Learning rate | 0.0001 |
| Epochs | 100 |
| Batch size | 100 |
| Loss function | Softmax |
| Sparse VAE | $\lambda_1 = 1$, $\lambda_0 = 10$ |
| $\beta$-VAE | [2, 4, 6, 8, 16] |
| VSC | $\alpha = 0.01$ |
| OI-VAE | $\lambda = 1, p = 5$ |
| Runtime per split | GPU, 1 hour |

PeerRead

| Settings | Value |
|---|---|
| # hidden layers | 3 |
| # layer dimension | 100 |
| Latent space dimension | 20 |
| Learning rate | 0.01 |
| Epochs | 40 |
| Batch size | 128 |
| Loss function | Softmax |
| Sparse VAE | $\lambda_1 = 0.001$, $\lambda_0 = 5$ |
| $\beta$-VAE | [2, 4, 6, 8, 16] |
| VSC | $\alpha = 0.01$ |
| OI-VAE | $\lambda = 1, p = 5$ |
| Runtime per split | GPU, 20 mins |

Semi-synthetic PeerRead

| Settings | Value |
|---|---|
| # hidden layers | 3 |
| # layer dimension | 50 |
| Latent space dimension | 20 |
| Learning rate | 0.01 |
| Epochs | 50 |
| Batch size | 128 |
| Loss function | Softmax |
| Sparse VAE | $\lambda_1 = 0.01$, $\lambda_0 = 5$ |
| $\beta$-VAE | [2, 4, 6, 8, 16] |
| VSC | $\alpha = 0.01$ |
| OI-VAE | $\lambda = 1, p = 5$ |
| Runtime per split | GPU, 30 mins |

Zeisel

| Settings | Value |
|---|---|
| # hidden layers | 3 |
| # layer dimension | 100 |
| Latent space dimension | 15 |
| Learning rate | 0.01 |
| Epochs | 100 |
| Batch size | 512 |
| Loss function | Gaussian |
| Sparse VAE | $\lambda_1 = 1$, $\lambda_0 = 10$ |
| VSC | $\alpha = 0.01$ |
| OI-VAE | $\lambda = 1, p = 5$ |
| Runtime per split | CPU, 15 mins |

