# OpenReview forum: "Identifiable Deep Generative Models via Sparse Decoding"
_TMLR — Accepted by TMLR_

### Review · Reviewer_KcCj · 2022-07-12

**Summary Of Contributions:**

The authors propose a new type of deep generative models (DGMs), called sparse DGMs, and prove analytically that under certain assumptions these models are identifiable. They then instantiate this model class with an amortized inference version, called sparse VAE, and show empirically that this model outperforms existing VAE models across a range of tasks.

**Broader Impact Concerns:**

There are no ethical concerns.

**Requested Changes:**

Overall, this is a really nice paper and I think it will be a nice contribution to this journal. However, I think there are a few improvements that could be made.

Critical:

- It is unclear to me how strong assumption A1 is, especially that the two anchor features have the exact same mapping $f_j$. Does this mean that the features essentially have to be copies of each other? How realistic is this assumption in the real data?
- As mentioned by the authors, their theory does not apply directly to the proposed sparse VAE, due to its non-convexity. It would be useful to discuss this discrepancy a bit more and give an intuition for how "close" the sparse VAE gets to the identifiable regime or what kind of adjustments in the model make it more or less identifiable.
- Relatedly, some of the design decisions are not motivated very clearly. For instance, how would the SSL prior work if the slab would not be a Laplace distribution? How would it compare to, e.g., a horseshoe prior? Some ablation studies would be useful to motivate why this particular model design is the best one.

Minor:

- Since the proposed sparse VAE is computationally more expensive than some of the baselines, it would be useful to report runtimes in the experiments to get a better intuition for how much additional compute one has to invest for the improvement in performance.

**Strengths And Weaknesses:**

Strengths:

- The paper is well written and easy to understand.
- Identifiability in DGMs is a very useful but elusive feature and this work provides a way to achieve it.
- The empirical performance of the proposed sparse VAE looks very promising.

Weaknesses:

- It is unclear how realistic some of the theoretical assumptions are.
- The connection between the theory and the presented sparse VAE model could be discussed in more detail.
- Some of the design choices are not fully motivated; ablation studies could help.

---

> ### Author Response · Authors · 2022-08-11
> **Response**
>
> We thank the reviewer for their time and comments. We respond to specific points below:
>
> > 1. It is unclear to me how strong assumption A1 is, especially that the two anchor features have the exact same mapping. Does this mean that the features essentially have to be copies of each other? How realistic is this assumption in the real data?
>
> In the synthetic data example, the anchor assumption does not exactly hold. Features $x_{\cdot 1}$ and $x_{\cdot 2}$ both have a linear dependence on $z_{\cdot 1}$ but have a different linear coefficient. (For the anchor assumption to hold exactly, $x_{\cdot 1}$ and $x_{\cdot 2}$ would need the same coefficient.) Similarly, features $x_{\cdot 4}$ and $x_{\cdot 5}$ have a different linear relationship to $z_{\cdot 2}$.
>
> Based on these results, we anticipate that the anchor assumption may be relaxed somewhat. Specifically, we conjecture that the anchor features require the same functional form in their dependence on $z$, but not necessarily the exact same function.
>
> Regarding real world data, it is possible that the anchor assumption holds in a variety of settings.  In particular, the anchor feature assumption is used in the topic modeling literature; we may expect that there are a few specialized words for each topic, and moreover that they have a similar mapping to the topic (e.g. synonyms). In movie ratings data, we may expect that movies from a single genre may be rated similarly (i.e. have a similar mapping). In gene expression data, genes corresponding to the same biological process may also have similar expression levels.
>
> > 2. As mentioned by the authors, their theory does not apply directly to the proposed sparse VAE, due to its non-convexity. It would be useful to discuss this discrepancy a bit more and give an intuition for how "close" the sparse VAE gets to the identifiable regime or what kind of adjustments in the model make it more or less identifiable.
>
> We have added a discussion of when we expect the sparse VAE to find the true factors (which exist due to identifiability).
>
> In particular, on page 8 we write:
>
> "An important implication of identifiability is consistency: if we learn the optimal parameters of the sparse VAE, then we will recover the true factors in the limit of infinite data. Specifically, if (i) the variational family $q_{\phi}(z|x)$ contains the true posterior $p_{\theta}(z|x)$ and (ii) the ELBO is maximized with respect to the parameters $\theta$, $\phi$ and $W$, then in the limit of infinite data, the sparse VAE will learn the true factors up to coordinate-wise transform and permutation."
>
> Note: this property is similar to Theorem 4 of Khemakhem et al. (2020) for their iVAE method.
>
> - Khemakhem, I., Kingma, D., Monti, R., & Hyvarinen, A. (2020). Variational autoencoders and nonlinear ICA: A unifying framework. In International Conference on Artificial Intelligence and Statistics (pp. 2207-2217). PMLR.
>
> > 3. Relatedly, some of the design decisions are not motivated very clearly. For instance, how would the SSL prior work if the slab would not be a Laplace distribution? How would it compare to, e.g., a horseshoe prior? Some ablation studies would be useful to motivate why this particular model design is the best one.
>
> We have added a comparison to the horseshoe prior in Appendix C.4.1. We find that the Spike and Slab Lasso prior has higher DCI than the horseshoe prior at higher levels of correlation between the factors. (Note: in Ročková and George (2018), the paper which introduces the Spike-and-Slab Lasso, the authors found that the Spike and Slab Lasso performs particularly well for correlated design matrices).
>
> We also have a comparison to the Lasso (where $\lambda_1=\lambda_0$) in Appendix C.4.1. Again, the Spike and Slab Lasso shows better performance.
>
> Finally, we note that a main contribution of the paper is that the form of sparsity we impose has implications for model identifiability. We found that the Spike-and-Slab Lasso performs well for this purpose but we do not claim it is exclusively always the best sparsity-inducing prior.
>
> To page 8, we have added: "The SSL prior is not the only sparsity-inducing prior that we could have used; however, we found it to work well empirically. We compare the SSL prior to the horseshoe (Carvalho et al. 2011) in Appendix C.4.1."
>
> - Veronika Ročková and Edward I George. The spike-and-slab lasso. Journal of the American Statistical Association, 113(521):431–444, 2018.
>
> > 4. Since the proposed sparse VAE is computationally more expensive than some of the baselines, it would be useful to report runtimes in the experiments to get a better intuition for how much additional compute one has to invest for the improvement in performance.
>
> We have reported runtimes in Table 6 of the appendix.

---

### Review · Reviewer_1jnL · 2022-07-21

**Summary Of Contributions:**

This paper introduces a new family of deep generative models that is identifiable by virtue of the anchor assumption (which states that there are at least two features which depend solely on that factor). Intuitively, this assumption effectively means that the observation serves as a noisy proxy for the latent variable itself and is what is used to guarantee identification of the model. The model is instatiated via a spike and slab
prior which is used to define a mask over which latent factors are used in the conditional distribution over the observations given the latent variables.

The model is evaluated on synthetic data, where it is found to recover the underlying sparsity of the generating process. In addition it is evaluated on several tabular datasets. On MovieLens (Recsys) it is found to improve upon recall and NDCG. On PeerRead, it is found to improve upon likelihood. On a data of mouse cortex cells, the latent representation learned by the model is found to improve predictive accuracy.


**Broader Impact Concerns:**

To the best of my knowledge, there are no broader impact concerns I anticipate with this manuscript.


**Requested Changes:**

* Running an evaluation on larger datasets
* Evaluate how the model performs as a function of feature size (and number of training examples)
* What happens in scenarios where there is model misspecification? i.e synthetic data where the anchor assumptions is violated?

**Strengths And Weaknesses:**

Strengths
* Clarity: I found the paper clear, well written and easy to follow
* Novelty: The idea to leverage the anchor assumption to obtain identifiability is neat and this work presents a compelling instantiation of this idea.

Weaknesses
* Evaluation: I think the biggest concern that I have is that method is solely evaluated on datasets that are relatively small (the largest number of features is ~500 and the largest number of training samples is ~100K). In order to push the limits of the method, its worthwhile to extend the evaluation to larger datasets.
   * The MovieLens dataset comes in a variety of different sizes, including the MovieLens20M which has in the tens of thousands of features. Given that this method is restricted to tabular data, I do think it is important for the work to evaluate on such larger training datasets.
   * It would be interesting to study the effect of increasing training/feature size. Specifically, this would enable practitioners to assess the utility of this method in different datasize/feature regimes.
   * Another reason to consider increasing the dimensionality of features is that tabular data often exhibits long-tails which can inhibit learning (e.g. https://arxiv.org/abs/1901.05534, https://arxiv.org/abs/1710.06085). This regime presents an interesting opportunity to assess the degree to which model restrictions that guarantee identifiability help/hurt learning on tabular data.

---

> ### Author Response · Authors · 2022-08-11
> **Response**
>
> We thank the reviewer for their time and comments. We respond to specific points below:
>
> > 1. Running an evaluation on larger datasets
> > 2. Evaluate how the model performs as a function of feature size (and number of training examples)
>
> Thank you for these suggestions. We are currently running a suite of experiments on a larger MovieLens dataset, as well as versions of MovieLens where we vary the number of training examples and feature size.
>
> Unfortunately, there have been issues with our cluster so we do not have the results yet -- we will have these results ready before August 22 (the reviewer recommendation deadline).
>
>
> > 3. What happens in scenarios where there is model misspecification? i.e synthetic data where the anchor assumptions is violated?
>
> In the synthetic data example, the anchor assumption actually does not exactly hold. Features $x_{\cdot 1}$ and $x_{\cdot 2}$ both have a linear dependence on $z_{\cdot 1}$ but have a different linear coefficient. (For the anchor assumption to hold exactly, $x_{\cdot 1}$ and $x_{\cdot 2}$ would need the same coefficient.) Similarly, features $x_{\cdot 4}$ and $x_{\cdot 5}$ have a different linear relationship to $z_{\cdot 2}$.
>
> Based on these results, we anticipate that the anchor assumption may be relaxed somewhat. Specifically, we conjecture that the anchor features require the same functional form in their dependence on $z$, but not necessarily the exact same function.

---

> > ### Author Response · Authors · 2022-08-23
> > **Added experiments**
> >
> > Update: we have added experiments to Appendix C.5 where we vary the dimension of the MovieLens dataset and compare the Sparse VAE to a VAE (the best performing methods on the MovieLens data). Generally, the Sparse VAE performs the same or better than the VAE in terms of negative heldout log likelihood and heldout recall.
> >
> > Note that unlike the VAE, the Sparse VAE also returns interpretable latent factors (Table 3).

---

### Review · Reviewer_TQnQ · 2022-07-25

**Summary Of Contributions:**

The paper proposes a method called Sparse VAE where the mapping between latent factors and observed features is enforced to be sparse in the sense that each observed feature depends only on a small number of latent factors. This is achieved by computing each observed feature independently using the same decoder but with a different subset of latent factors as input (obtained via a mask on the full set of latent factors). The mask varies across output features but is the same across data points, which the authors motivate using a genomics application where each observed feature always corresponds to the same gene. The authors further motivate the approach through an identifiability theoretical analysis, highlighting that compared to other DGMs the proposed approach is identifiable. Finally the authors demonstrate the benefits of the approach on a number of synthetic and real tasks.

**Broader Impact Concerns:**

No concerns.

**Requested Changes:**

Given that identifiability has been analyzed in the context of NMF, I would consider either including NMF in Figure 3 (identifiability) or describing why one cannot use NMF in that context. In general, given the tabular data assumption, I think that the stronger the discussion (and empirical comparison) with alternative methods tailored for tabular data like movie reviews, the stronger the paper.

Overall though, the paper is well written with all the strengths and weaknesses discussed clearly. The paper describes how the identifiability analysis motivates the modeling choices (e.g. highlighting that the SSL prior is likely to yield the sparse mapping that satisfies the anchor assumption required for identifiability) and there is at least one (synthetic) experiment to illustrate each claim e.g. identifiability, interpretability, and improved held out performance on real data. Therefore my recommendations above are not critical for securing my recommendation for acceptance.

**Strengths And Weaknesses:**

Strengths: The paper is well written and contributes to the DGM literature with a new modeling approach, theoretical analysis, and experiments designed to illustrate the proposed approach's main benefits.

Weakness:
* While the approach has some advantages over DGMs (e.g. identifiability) it has more limited applicability (can only be applied to tabular data, i.e. where the number of observed features is fixed and where each observed feature has the same interpretation across data points).
* Compared to the generic VAE, the proposed approach is also more computationally expensive.

---

> ### Author Response · Authors · 2022-08-11
> **Response**
>
> We thank the reviewer for their time and comments. We respond to specific points below:
>
> > While the approach has some advantages over DGMs (e.g. identifiability) it has more limited applicability (can only be applied to tabular data, i.e. where the number of observed features is fixed and where each observed feature has the same interpretation across data points).
>
> > Compared to the generic VAE, the proposed approach is also more computationally expensive.
>
> Both of these limitations are correct - we hope we have been forthcoming about this limitation, and have reported run times in the appendix (page 23).
>
> > 1. Given that identifiability has been analyzed in the context of NMF, I would consider either including NMF in Figure 3 (identifiability) or describing why one cannot use NMF in that context. In general, given the tabular data assumption, I think that the stronger the discussion (and empirical comparison) with alternative methods tailored for tabular data like movie reviews, the stronger the paper.
>
> Thank you for your comment. As NMF is a linear method, it cannot reconstruct the nonlinear data generation process using only two latent dimensions. We have added this discussion to the paper.

---

### Author Response · Authors · 2022-08-11
**Overview of response**

We thank the reviewers for their helpful and thoughtful comments. We are encouraged that the reviewers think:

- the paper is "well written" and "contributes to the DGM literature with a new modeling approach" (Reviewer TQnQ)
- that "the idea to leverage the anchor assumption to obtain identifiability is neat and this work presents a compelling instantiation of this idea." (Reviewer 1jnL)
- this is a "well-written, nice paper" that would be "a nice contribution to this journal" (Reviewer KcCj)

We respond to reviewers specific comments below.

---

### Decision · Action_Editors · 2022-09-09

**Recommendation:** Accept as is

**Comment:**

The paper introduces sparse VAE, a deep generative model applicable when each observed dimension is dependent on a potentially different small subset of latent variables. This sparse dependence is modelled by applying a different learned sparse mask to the latent vector for each observed dimension. The authors prove that the resulting model is identifiable under certain assumptions and show that it performs well in practice. The paper is well written and makes progress in a difficult area. The authors have addressed most of the reviewer concerns with their rebuttal.